# Au-Based Nanoparticles Enhance Low Temperature Tolerance in Wheat by Regulating Some Physiological Parameters and Gene Expression

**DOI:** 10.3390/plants13091261

**Published:** 2024-04-30

**Authors:** Yuliya Venzhik, Alexander Deryabin, Kseniya Zhukova

**Affiliations:** K.A. Timiryazev Institute of Plant Physiology, Russian Academy of Sciences, 127276 Moscow, Russia; anderyabin@ifr.moscow (A.D.); zhukova@ifr.moscow (K.Z.)

**Keywords:** Au-based nanoparticles, low temperature tolerance, lipid peroxidation, photosynthetic pigments, sugars, *Cor-*genes, *Triticum aestivum*

## Abstract

One of the key problems of biology is how plants adapt to unfavorable conditions, such as low temperatures. A special focus is placed on finding ways to increase tolerance in important agricultural crops like wheat. Au-based nanoparticles (Au-NPs) have been employed extensively in this area in recent years. Au-NPs can be produced fast and easily using low-cost chemical reagents. When employed in microdoses, Au-NPs are often non-toxic to plants, animals, and people. In addition, Au-NPs mainly have favorable impacts on plants. In this study, we investigated the effect of Au-NP seed nanopriming (diameter 15.3 nm, Au concentration 5–50 µg mL^−1^) on cold tolerance, as well as some physiological, biochemical and molecular parameters, of cold-sustainable wheat (*Triticum aestivum* L.) genotype Zlata. The treatment with Au-NPs improved tolerance to low temperatures in control conditions and after cold hardening. Au-NPs treatment boosted the intensity of growth processes, the quantity of photosynthetic pigments, sucrose in leaves, and the expressions of encoded RuBisCo and *Wcor15* genes. The potential mechanisms of Au-NPs’ influence on the cold tolerance of wheat varieties were considered.

## 1. Introduction

One of the main goals of biology is the study of the physiological, biochemical, and molecular mechanisms of plants’ adaptation to abiotic factors. The majority of arable land is currently located in an unstable agricultural region. This makes it relevant to study the adaptation strategies of such an important crop as wheat to low temperatures. Wheat is one of the cold- and freezing-tolerant crops that is able to survive even near-zero negative temperatures. The cultivation of wheat for centuries has resulted in the development of a wide range of cold- and freezing-tolerant varieties, and cold-sustainable genotypes respond differently to low temperatures [1]. No doubt, the diversity of wheat genotypes and the characteristics of their stress response make this crop an ideal subject for the research of physiological, biochemical, and molecular mechanisms that contribute to improving of their tolerance [2,3].

At low temperatures, cold-tolerant genotypes of wheat are able to maintain a stable content of photosynthetic pigments, ribulose-1,5-bisphosphate carboxylase/oxygenase (RuBisCo) activity, and photosynthetic intensity, and as a result, accumulate sugars required for the adaptation to low temperature [4,5,6]. It is known that more tolerant wheat genotypes at low temperatures enact a powerful response, including the expression of a number of *Cor-*genes (Cold-Regulated Genes) [7,8]. As for cold-sustainable genotypes, their set of mechanisms for maintaining cold and freezing tolerance may be significantly limited [4,5,9].

Many *Cor-*genes encode COR proteins, which are important in plant adaptation to low temperatures. These are hydrophilic low-molecular-weight proteins that function as chaperones in cells, limiting protein aggregation, cell water loss, and membrane degradation [10]. The WCS (Wheat Cold Specific) family is also referred to as COR proteins—dehydrins [7,11]. The molecular weights of these proteins encoded by the *Wcs120* gene family range from 12 to 200 kD [8]. The accumulation of dehydrins and increased expression of genes encoding them are associated with the development of tolerance to low temperatures [7,12].

Plants also have cold-sensitive genes that encode chloroplast-targeted proteins [8]. These include the cold-sensitive wheat *Wcor15* gene [13], which is related to *Arabidopsis cor15* gene [14,15]. It encodes proteins that form an ordered structure when exposed to low temperature stress. These proteins can bind to chloroplast membranes in this condition, making them more tolerant [16].

The use of metal nanoparticles (NPs), which can have a positive effect on plant metabolism and stress tolerance in low concentrations, is a promising trend [17,18,19,20]. NPs cross cellular barriers and affect practically all processes in the plant organism due to their small size (less than 100 nm) and unique physical, electrical, optical, and chemical properties [17,18,21,22]. NPs are being employed more and more widely in biology, medicine, and agriculture as plant growth and development regulators, medicinal components, herbicidal/pesticidal chemicals, and nanofertilizers [19,21,23].

It is known that the effects of NPs on plants depend on a number of factors (for example, type of NPs, method of treatment, plant species, experimental conditions, etc.). First of all, the type of nanoparticles matters. In our research, we used Au-based NPs (Au-NPs). We should emphasize that Au-NPs occupy a special place among metallic NPs. On the one hand, Au-NPs can be produced fast and easily using low-cost chemical reagents [24,25,26]. On the other hand, when employed in microdoses up to 100 µg mL^−1^, Au-NPs are often non-toxic to plants, animals, and people [27,28,29,30]. According to the literature, Au-NPs mainly have favorable impacts on plants [26,29,30]. The increase in seed germination rate and growth intensity under the influence of Au-NPs was shown in maize [31], *Arabidopsis* [32], mungbean [33], mustard [34,35], gloriose [36], arugula [37], lavender [38], watermelon [39], onion [40] and barley [41]. Under the influence of Au-NPs, researchers observed increased photosynthetic intensity [33,42], and changes in fluorescence parameters [43], photosynthetic pigment content [31,33,34] and chloroplast ultrastructure [31]. The presence of unique optical properties associated with the excitation of localized plasmon resonances during light interaction is responsible for the worldwide scientific interest in the use of Au-NPs in biology and medicine as photoprotective substances that reduce the risks of oxidative stress [24,28,29,44,45,46]. In this regard, Au-NPs have recently attracted a lot of attention as substances with the potential to increase crop environmental tolerance. It is worth noting that there are a few studies on the impact of Au-NPs on plant metabolism and tolerance. According to studies in barley and *Arabidopsis*, Au-NPs that enter plants through the roots alter the expression of *DGR1* and *DGR2* genes that code cell wall proteins [29], as well as changing chemical composition and increasing the stiffness of the cell wall, which helps plants withstand stress [41]. The treatment of rice plant roots with nanocomposites containing Au-NPs reduced cadmium toxicity by limiting root cell uptake and lowering the risk of oxidative stress [47,48]. Under saline conditions, Au-NPs preserved ionic balance, reduced the formation of NO as a signaling molecule and regulator of many processes in the plant, and boosted the activity of antioxidant system enzymes [49]. The potential for further integrating Au-NPs into biological sciences and agriculture is quite intriguing in this regard. It is critical to research them as substances that improve plant tolerance to low temperatures. There are no such studies in the literature, in connection with which our research is groundbreaking.

In a previous study [50], we demonstrated for the first time that Au-NPs can increase the tolerance to low temperatures of the freezing-tolerant genotype of wheat Moskovskaya 39. However, it is an open question whether Au-NPs can increase the low temperature tolerance of more cold-sensitive genotypes. In this research, we explored the effects of Au-NPs on the low temperature tolerance of cold-sustainable wheat genotype Zlata. We chose nanopriming (soaking seeds in NPs solutions) as the mode of nanoparticle treatment. Nanopriming is largely regarded as the safest, simplest, and most cost-effective method of treating plants with NPs. This allows for scientific and agricultural applications. We attempted to discover potential mechanisms by which Au-NPs can influence plant stress tolerance. For this purpose, we assayed the following indices: survival of wheat seedlings after freezing, seed germination, length of the first leaf, the content of malondialdehyde (MDA), photosynthetic pigments and soluble sugars in leaves and the expression of some genes of photosynthetic apparatus (PSA) and *Cor-*genes, which are important in plant acclimation to low temperatures.

## 2. Results

It was necessary to assess the impacts of various Au-NPs concentrations (5, 10, 20 and 50 µg mL^−1^) on the low temperature tolerance of wheat genotype Zlata at the start of the study. The survival rate of wheat plants at 0 °C was 100% in all experiment variants. The survival rate of unhardened seedlings at −3 °C was increased by a factor of two to three after exposure to Au-NPs, but only at concentrations of 5 and 10 µg mL^−1^ (Table 1). When we dropped the temperature to −5 °C, all the plants died. The tolerance to cold was improved by low-temperature hardening and was shown to be 60%. The stimulating impact of Au-NPs on cold tolerance was clearly established at −5 °C. There, Au-NPs at a concentration of 10 µg mL^−1^ increased cold tolerance from 7% to 67% (Table 1).

We assessed the impacts of various Au-NPs concentrations (5, 10, 20 and 50 µg mL^−1^) on the seed germination and growth of the first leaf of wheat seedlings (Figure 1). The stimulating impact of Au-NPs on seed germination was only noticed at a concentration of 20 µg mL^−1^ (Figure 1A). At the same time, Au-NPs treatment accelerated the growth of the first leaf of seedlings at all concentrations tested (Figure 1B). Under hardening, the growth of seedlings was completely stopped (Figure 1C).

Based on the results of concentration tests, a Au-NPs concentration of 10 µg mL^−1^ was chosen as causing the maximum effect on the low temperature tolerance of unhardened and hardened seedlings of wheat genotype Zlata (Figure 2 and Figure 3).

Table 2 displays data on the Au contents in roots, seeds and leaves of wheat. The study revealed that Au is present in Au-NPs-treated wheat roots and seeds. Furthermore, a trace of Au was discovered in the wheat leaves.

We studied MDA accumulation in wheat tissues as one of the end products of lipid peroxidation (LPO) of cell membranes. Treatment with Au-NPs both in control and hardening conditions did not affect MDA content in wheat leaves (Table 3).

Under control conditions, Au-NPs treatment resulted in a significant increase in total chlorophyll content without affecting carotenoids concentration or pigment ratio in wheat leaves (Figure 4A,C,E,G,I). Au-NPs treatment had no influence on photosynthetic pigment quantity or ratio under low-temperature conditions (Figure 4B,D,F,H,J).

Au-NPs treatment had no effect on the buildup of leaf dry weight of wheat leaves under control conditions. Dry leaf weight rose under hardening conditions, and Au-NPs treatment accelerated this process (Table 3).

Au-NPs treatment under control conditions considerably lowered the sugar content in wheat leaves by decreasing the content of monosaccharides (glucose and fructose). The content of sugars in leaves did not change under low-temperature conditions, whereas Au-NPs treatment increased them due to enhanced sucrose accumulation (Table 3).

We analyzed the effects of Au-NPs on the expression level of PSA genes *RbcS* and *RbcL*, encoding, respectively, the small and large subunits of RuBisCo. It was found that the expression level of *RbcS* and *RbcL* was increased in Au-NPs-treated plants under control conditions (Figure 5A,C). Under low-temperature hardening, Au-NPs had no significant effect on the expression levels of these genes (Figure 5B,D).

We also examined the impacts of Au-NPs on the expression of *Wcor726* and *Wcor15*, encoding protective proteins. Expression of the *Wcor15* gene increased 8-fold in wheat leaves, whereas the expression of the *Wcor726* gene did not change (Figure 6A,C). Au-NPs treatment also slightly enhanced *Wcor15* gene expression under low-temperature conditions (Figure 6B,D).

## 3. Discussion

In our experiments, wheat seeds of cold-susceptible genotype Zlata were soaked in Au-NPs solutions (5, 10, 20, and 50 µg mL^−1^) for 24 h. Based on data from the literature, concentrations of Au-NPs solutions below 50 μg mL^−1^ generally have a stimulating effect on plants [28,29]. The nanopriming procedure, which involves soaking seeds in NPs solution, is thought to be the mildest for such studies [31,39,51]. In this situation, NPs most likely enter the seeds passively with water, as well as through the damaged sections of the seed coat [52]. The mechanism of NPs continuing “work” in the plant organism is not entirely clear. Au-NPs are not only adsorbed and/or deposited in the treated seeds, but also penetrate further into the seedlings, where they spread through their tissues and cells. For instance, Au-NPs were found in the leaves of wheat, oat, and maize seedlings developed from seeds treated with Au-NPs, as demonstrated by fluorescence microscopy and optical emission spectroscopy [31,53]. Employing plasma-atomic emission and inductively coupled plasma mass spectrometry, we found Au in roots and seeds of treated seedlings. Minor concentrations of Au were found in the leaves of wheat grown from Au-NPs-treated seeds (Table 2). It is important to note that these plants differed from the control (untreated) in a number of indicators and, most critically, had a different adaptive status due to their ability to develop higher tolerance to low temperatures.

The Au-NPs concentration tests revealed that the nanopriming of wheat improved its tolerance to low temperatures in control conditions and after cold-hardening (4 °C, 7 d). The maximum effect was achieved in genotype Zlata after the treatment of seeds with Au-NPs at a concentration of 10 μg mL^−1^ (Table 1). Additionally, the treatment of seeds with Au-NPs induced the growth of plants (Figure 1B). We selected a concentration of 10 μg mL^−1^ for the further study of Au-NPs’ effects on physiological, biochemical and molecular parameters of wheat.

Similar results on the growth-promoting effects of low concentrations of Au-NPs were obtained in many other plant species, including maize, *Arabidopsis*, mungbean, mustard, gloriosa, arugula, lavender, and watermelon [31,32,34,35,38,40,41]. It is hypothesized that Au-NPs may participate in cell differentiation through molecular pathways [54]; for example, there is evidence of the ability of Au-NPs to activate genes that are involved in the regulation of aquaporins and responsible for the cell cycle [29,41]. Additionally, Au-NPs can affect cell differentiation through miRNAs—short, non-coding RNAs that regulate a variety of biological functions, including growth and stress responses. Thus, Au-NPs altered the expressions of miR398, miR408, miR164, miR167, and miR169 in *Arabidopsis* [32]. Furthermore, miR167 expression was linked to the activity of genes that regulate plant reproductive processes via affecting auxin signal transduction pathways. The altered expressions of miR169, miR398 and miR408 influenced the sizes of seedlings, as well as the development of their root systems, and was responsible for the early flowering of plants and hastened seed maturation [32].

We emphasize that the effects of Au-NP on PSA were expressed via a significant increase in the level of chlorophylls in the leaves of wheat under control conditions (Figure 4A). This could imply that the PSA is actively working, and that thylakoid membranes are stable. Another indication of the PSA’s stability is the consistent ratio of photosynthetic pigments of wheat under the impact of Au-NPs (Figure 4E–H). Other studies indicate that Au-NPs increased chlorophyll content in soybean, corn, and mustard plants [31,33,34]. Researchers observed an increase in the electronic transport rate and Hill reaction [33,43], as well as changes in fluorescence parameters and photosynthetic intensity [33,42], under the influence of Au-NPs. The peculiar features of Au-NPs in this area are attributed to the effect of plasmon resonance. The impact is that electrons on the surfaces of Au-NPs can dramatically enhance their activity due to their propensity to collectively oscillate in response to a certain wavelength of light (Hu and Xianyu 2021) [45]. Active electrons on the surfaces of Au-NPs can “trap” photons of light, facilitating the transfer of energy in the light-harvesting complex (LHC) [33,43]. Au-NPs’ effects on PSA parameters were shown not only in leaves sprayed with Au-NPs, but also in plants grown from treated seeds [55].

NPs also change the intensity of photosynthetic pigment synthesis and the activity of PSA by regulating the expression of genes involved. For example, zinc oxide NPs increased the expression of genes encoding photosystem structural units (*PSAD2, PSAE2, PSAK*) and photosynthetic pigment synthesis genes (*CAO, CHLG, GGPS6, PSY, PDS, ZDS*) [56]. Silicon oxide NPs enhanced the expression of *PsbH, PsbB*, and *PsbD* genes encoding photosystem II (PS II) proteins [57]. Titanium oxide particles enhanced the expression of genes involved in the synthesis of RuBisCo [58]. Au-NPs’ effects on *RbcL* and *RbcS* genes were shown in our study under control conditions (Figure 5A,C). The expression levels of RbcL and RbcS genes encoding the large and small subunits of RuBisCo were increased upon Au-NPs treatment, which may indicate an increase in RuBisCo as a key PSA enzyme.

Under low-temperature hardening, Au-NPs caused an additional increase in the cold tolerance of wheat seedlings (Table 1). However, the effects of Au-NPs on growth processes and PSA activity were almost completely cancelled out by the effect of low temperature. At the same time, under the Au-NPs treatment, the process of increasing cold tolerance was accompanied by a significant increase in sucrose content (Table 3). The buildup of soluble sugars is known to be the most important nonspecific plant response to abiotic stress, which is required for survival under low temperatures [59,60]. Frost tolerance (resistance to sub-zero temperatures) in cereals is related to the outflow of water from cells into intercellular spaces, and the formation of intercellular ice. At the same time, the high content of sugars in the cells reduces the risks of dehydration and decreases ice nucleation temperature. Sugars act as osmolytes and cryoprotectants; they stabilize membranes and act as antioxidants [61,62]. Furthermore, sugars are reserve substances, and their accumulation is required for the energy-intensive cold adaptation process [63,64,65].

It is understood that Au-NPs, as alien exogenous particles to the plant organism, can operate as triggers for oxidative stress [66]. The development of LPO products, on the other hand, is a critical link in the chain of plant cell-adaptive responses to abiotic stress [59]. The products of LPO (including reactive oxygen species, MDA and ET) created in plant cells during oxidative stress development can induce the ICE-CBF/DREB1 ((ICE)-C-Repeat Binding Factor/Dehydration Responsive Element Binding Factor 1) signaling pathway of cold adaptation in plants, which triggers the expression of *Cor-*genes [60,67]. Additionally, they stimulate the antioxidant system (AOS), which in plants is the most significant non-specific protective enzymatic mechanism. It was shown in watermelon and mustard that exposure to Au-NPs was accompanied not only by an increase in MDA in leaves, but also by an increase in the activity of AOS enzymes [35,39]. We found that under either control or hardening conditions, the Au-NPs treatment had no effect on the MDA contents of wheat leaves (Table 3). It can be concluded that in our case, NPs do not cause oxidative stress.

We studied the effects of Au-NPs on the expression of the *Cor*-genes *Wcor15* and *Wcor726*. Note that the *Wcor726* gene belongs to the *Wcs120* family, which is specific to wheat [68,69]. These genes are regulated by low temperatures and encode WCS protein accumulation, which directly correlates with the ability of plants to increase tolerance to low temperatures [68]. According to some research [7], *Wcor726* is a member of the family D11 of the large class LEA (Late Embryogenesis Abundant) proteins, which are dehydrins involved in plant defense against a variety of stresses.

WCOR15 appears to belong to group III LEA proteins that target chloroplasts [70,71]. The *Wcor15* gene encoding chloroplast proteins contains CRT/DRE cis-elements similar to Wcs120 [13]. There is proof that the *Wcor15* gene produces the WCOR15 protein, which builds up in chloroplasts and reduces the degree of photoinhibition of PS II [72,73].

According to our studies, under the influence of Au-NPs, in control conditions an 8-fold increase in the expression of the *Wcor15* gene was observed (Figure 6C). Under low-temperature hardening, the expressions of *Wcor726* and *Wcor15* in untreated wheat plants were significantly increased. Au-NPs treatment caused a small additional increase in the expressions of these genes. As shown in other studies, plants of cold-sensitive wheat genotypes tended to accumulate less *Wcor15* transcripts [13]. However, our data suggest that in wheat genotype Zlata, the increase in freezing tolerance under the influence of Au-NPs seems to be related to the expression of this gene. Most likely, this ensured an increase in their freezing tolerance.

This research demonstrates that the nanopriming method can successfully deliver Au-NPs to seeds, and the NPs then permeate the plant organism. Plants grown from treated seeds differed from control (untreated) plants not only in physiological, biochemical and molecular characteristics, but also in adaptive status. The treatment with Au-NPs improved tolerance to low temperatures in control conditions and after cold hardening. Au-NPs treatment boosted the intensity of growth processes, the quantity of photosynthetic pigments, sucrose in leaves, and the expression of genes, such as encoded RuBisCo and the *Wcor15* gene (Figure 7). The mechanisms of cold tolerance enhancement under the influence of Au-NPs are not completely clear. It can be assumed based on our data that improvements in wheat’s low temperature tolerance are associated with a high content of soluble sugars and an increased level of Wcor15 gene expression.

Comparing biochemical and molecular processes related to this phenomenon in the freezing-tolerant genotype Moskovskaya 39 (presented in previous study [50]) and the cold-susceptible genotype Zlata revealed that, under control conditions, Au-NPs treatment increased the content of photosynthetic pigments and the expression of the *Wcor15* gene in both wheat genotypes. There were also differences between genotypes. In the freezing-tolerant genotype Moskovskaya 39, the rise in cold tolerance was followed by an increase in peroxide processes, but in cold-susceptible genotype Zlata, the increase in peroxide processes was not detected, while the concentrations of suitable osmolytes in tissues rose. The considerable improvement in cold tolerance under Au-NPs treatment in freezing-tolerant genotype Moskovskaya 39 under low-temperature hardening conditions was related to a complex of adaptation changes: increased chlorophyll and carotenoids content, sucrose, and the expression of *Wcor15* and *Wcor726* genes. The increase in cold tolerance of cold-susceptible genotype Zlata under these conditions was connected with the expression of the *Wcor15* gene and a rise in multiple sugars.

## 4. Materials and Methods

### 4.1. Experimental Design

The experiments were carried out according to the following scheme (Figure 8). The first stage is the chemical synthesis of Au-NPs solutions. Based on data from the literature, we selected four concentrations of Au-NPs solutions—5, 10, 20 and 50 μg mL^−1^—since it is known that concentrations of Au-NPs solutions below 50 μg mL^−1^ generally have a stimulating effect on plants [28,29].

The second step was the nanopriming (soaking for 24 h) of seeds in Au-NPs solutions at the selected concentrations. The next step involved growing plants from treated seeds under controlled conditions for 10 days.

Then, we determined the tolerance of plants to low temperatures (concentration tests) using growth indicators and survival rates after freezing. As a result, the concentrations of Au-NPs solutions with the maximum effect on low temperature tolerance were selected.

All further experiments were conducted under control conditions (at 22 °C) and after cold hardening (4 °C, 7 d) using Au-NPs concentrations giving the maximum effect on the low temperature tolerance of plants.

### 4.2. The synthesis of Au-NPs

Gold spherical nanoparticles were created using the citrate method [25] by reducing chloroauric acid (Sigma-Aldrich, St. Louis, MO, USA) with sodium citrate (Fluka, St. Gallen, Switzerland). The reduction was accomplished by heating 250 mL of a 0.01% aqueous solution of chloroauric acid to 100 °C in an Erlenmeyer flask on a magnetic stirrer with a reflux water condenser. In the next step, 7.75 mL of 1% aqueous sodium citrate solution was added, and the mixture was boiled for an additional 30 min until a crimson sol formed. Freshly produced Au-NP solutions were transferred to sterile glass vials with tight-fitting lids and kept at 4 °C.

The resulting Au-NPs were studied by transmission electron microscopy, spectroscopy, and dynamic light scattering [74]. According to the measurement results, the average diameter of Au-NPs was 15.3 nm. Detailed characterizations and TEM images of Au-NPs have been presented in our previous study [50].

### 4.3. Growth Conditions

Seeds of cold-susceptible wheat (*Triticum aestivum* L., Poaceae) genotype Zlata (Federal Research Center “Nemchinovka”, Moscow, Russia) were used in the investigations. The seeds were immersed in Au-NPs solutions for 24 h, and then were washed in distilled water. The seeds were germinated in distilled water at 22 °C, 60–70% relative humidity, and a photoperiod of 16 h (illumination 100 μmol photons m^−2^ s^−1^) with OSRAM L 80W/640 lamps (Osram, Smolensk, Russia). The seedlings were grown on distilled water under the same conditions until 10 days of age. Then, a part of plants was hardened at 4 °C for 7 days in a KBW-240 climatic chamber (Binder, Tuttlingen, Germany).

### 4.4. Parameters of Growth

Wheat seed germination was measured on day 7 and reported as a percentage. The initial leaf’s length was measured both at 10 days of growth and at 17 days following hardening. In each variant of the treatment, 30 plants were used in 3 replications (90 plants in all). The experiment was repeated 2 times.

### 4.5. Survival of Plants after Freezing

In order to identify the concentrations of Au-NPs that have the maximum protective effect on wheat plants of genotype Zlata, the degree of tolerance to low temperatures was determined by direct freezing. The rate of survival of seedlings was measured after freezing in a climatic chamber MIR-153 (Sanyo, Osaka, Japan) at 0 °C, −3 °C, −5 °C, −7 °C and −9 °C, with a 24 h interval. After freezing, the plants were preserved for one day at 4 °C in the dark before being transplanted to normal temperatures (22 °C, daylight) for 72 h. The number of undamaged seedlings as a percentage of the total number of frozen plants was used to calculate the survival rate. The choice of temperatures for freezing was made on the basis of our own research and literature data. In each variant of the treatment, 30 plants were used in 3 replications. The experiment was repeated 2 times.

### 4.6. Quantification of Au

We analyzed tissues of 10-day-old seedlings grown at 22 °C from seeds treated with Au-NPs at a concentration of 10 μg mL^−1^. Roots, seeds, and leaves of wheat seedlings were oven-dried at 70 °C for 72 h, weighed, and digested (the seeds were first cleaned and dried). The digestion was performed using microwave-assisted digestion (UltraClave III, Milestone, Santa Clara, CA, USA) at 105 °C for 15 min with 3 mL of plasma pure HNO_3_. Then, the volume of the sample was adjusted to 10 mL using deionized water. The concentration of Au was measured using an inductively coupled plasma-atomic emission spectrometer (Agilent Technologies, Santa Clara, CA, USA) and an inductively coupled plasma mass spectrometer 820 (Bruker, Bremen, Germany). In each variant of the treatment, 3 statistical repetitions were carried out. The experiment was repeated 2 times.

### 4.7. Dry Matter Content

Given that plants lose water at low temperatures, the data for all indicators were converted to dry weight. The dry matter content was determined by drying them in a thermostat (at 100–105 °C) to a constant weight and expressed as a percentage of the sample’s initial wet weight.

### 4.8. Lipid Peroxidation Level (LPO)

LPO level was evaluated as the content of MDA [75] with slight modifications. Leaf samples (300 mg) were homogenized in 5 mL of extraction medium (0.35 M NaCl in 0.1 M Tris-HCl buffer, pH 7.6). The homogenate (3 mL) mixed with 2 mL 0.5% thiobarbituric acid in 20% trichloroacetic acid was incubated (95 °C for 30 min), then cooled and filtered. The extraction medium with the reagent was employed as a control. Results were adjusted for nonspecific absorbance by subtracting the values observed at 532 and 600 nm. A Genesys 10UV spectrophotometer (Thermo Electron Corporation, Waltham, MA, USA) was used in the study. MDA concentration was determined using a molar extinction coefficient (ε = 1.56·10^5^ M^−1^ cm^−1^) in μM g^−1^ dry weight of leaves. In each variant of the treatment, 3 statistical repetitions were carried out. The experiment was repeated 3 times.

### 4.9. Content of Chlorophylls and Carotenoids

Chlorophyll *a* (Chl *a*), chlorophyll *b* (Chl *b*), and carotenoids (Car) concentration in the leaves were measured spectrophotometrically at wavelengths 663, 646, and 470 nm, respectively, in an 80% acetone solution. The contents of pigments were calculated using the formula [76]:C_car_ = (1000*D*_470_ − 3.27C_a_ − 104C_b_)/198, C_a_ = 12.21*D*_663_ − 2.81*D*_646_, C_b_ = 20.13*D*_646_ − 5.03*D*_663._

Total chlorophyll was determined as the sum of chlorophylls *a* and *b*. The pigment content was expressed as mg g^−1^ dry weight of leaves.

We calculated chlorophyll portion in light harvesting complex (LHC), assuming that practically all Chl *b* was located in LHC and the chlorophyll ratio Chl *a*/Chl *b* in this complex was equal to 1.2 [77]: LHC (%) = (Chl *b* + 1.2 × Chl *b*)/(Chl *a* + Chl *b*) × 100.

In each variant of the treatment, 3 statistical repetitions were carried out. The experiment was repeated 3 times.

### 4.10. Soluble Sugars (Glucose, Fructose, and Sucrose) Content

Solution of sugars were obtained from 500 mg of plant tissue after ethanol extraction three times. The fructose content was evaluated via the interaction of ketoses with resorcinol, and then the sucrose concentration was recalculated [78]. The content of glucose was measured by the glucose oxidase method using the Olvex diagnosticum Kit (Vital Diagnostics, Saint Petersburg, Russia). In each variant of the treatment, 3 statistical repetitions were carried out. The experiment was repeated 3 times.

### 4.11. Total RNA Extraction and cDNA Synthesis

Total RNA was isolated from 50 mg of leaf tissue samples using the Spectrum Plant Total RNA Kit (Sigma-Aldrich, USA) according to the manufacturer’s instructions with modification of the homogenization step according to [79]. The quality and quantity of purified RNA was determined by the NanoDrop-2000 spectrophotometer (Thermo Fisher Scientific, Waltham, MA, USA) and analyzed by 2% agarose gel electrophoresis. To remove residual genomic DNA impurities, total RNA preparations were treated with DNase I (Thermo Fisher Scientific, USA).

cDNA was synthesized using a RevertAid reverse transcription kit (Thermo Fisher Scientific, USA).

### 4.12. Gene Expression by RT-qPCR

Real-time quantitative PCR (RT-qPCR) was carried out on the CFX96 Touch™ (Bio-Rad, Hercules, CA, USA), using the SYBR Green I intercalating dye (Evrogen, Moscow, Russia). The reaction mixture for quantitative PCR in a volume of 25 μL contained 5 μL of qPCRmix HS SYBR (Evrogen), 0.2 μM of each primer, and 15 ng of the cDNA template. The following amplification conditions were used: 95 °C for 5 min, followed by 40 cycles of 95 °C for 15 s, 60 °C for 30 s, and 72 °C for 30 s.

Gene-specific primers (Table 4) for the amplification of large and small subunits of RuBisCo (RbcS and RbcS) target genes were borrowed from [80]; gene-specific primers for amplification of the target genes (*Wcor726*, *Wcor15*) and reference genes (*TaAct7*, *TaRP15*) were selected using the Primer-BLAST (https://www.ncbi.nlm.nih.gov/tools/primer-blast/, accessed on 13 March 2024) and OligoAnalyzer™ Tool (https://eu.idtdna.com/pages/tools/oligoanalyzer, accessed on 13 March 2024) online resources. Primers were designed to bind to all three wheat sub-genomes [80].

*Wcor726* encodes WCOR 726 dehydrin, which refers to Wheat Cold Specific family proteins that are important in plant cold and freezing adaptation. *Wcor15* encodes the WCOR15 chloroplast-targeted COR protein. The transcript levels were normalized to the expression of the reference genes *TaAct7* (actin) [81] and *TaRP15* (RNA polymerase I, II, and III, 15 kDa subunit) [82]. The relative expression levels were calculated using the Pfaffl method [83]. Each qRT-PCR reaction was performed in three biological and two technical replicates.

### 4.13. Statistical Analysis

Subject to the method used, the experiments were repeated three times to obtain similar reproducible results. Statistical significance was calculated via one-way ANOVA with a Tukey’s test (*p* < 0.05) using Origin 7.0 software. The tables and figures show the mean values and their standard errors.

## 5. Conclusions

Our results demonstrate that Au-NPs can “reprogram” the plant organism by altering the metabolism and the expression of genes involved in stress responses. It is important that Au-NPs act not only on freezing-tolerant, plants as we have previously established [50], but also on varieties whose genetic tolerance to low temperatures is reduced. As a result, plants are able to withstand low temperatures much better. Au-NPs regulated growth processes, increased the content of photosynthetic pigments and soluble sugars in leaves, and also increased the expression of *COR* genes and PSA genes. It should be mentioned that more research in this area is needed to better understand the effects of Au-NPs on plants, and the mechanisms of action in the plant organism. Perhaps in the future, Au-NPs can be used as growth and development stimulants as well as potential adaptogens. The development of strategies for the use of NPs should aim to reduce agricultural risks, i.e., focus on the development of sustainable agriculture.

## Figures and Tables

**Figure 1 plants-13-01261-f001:**
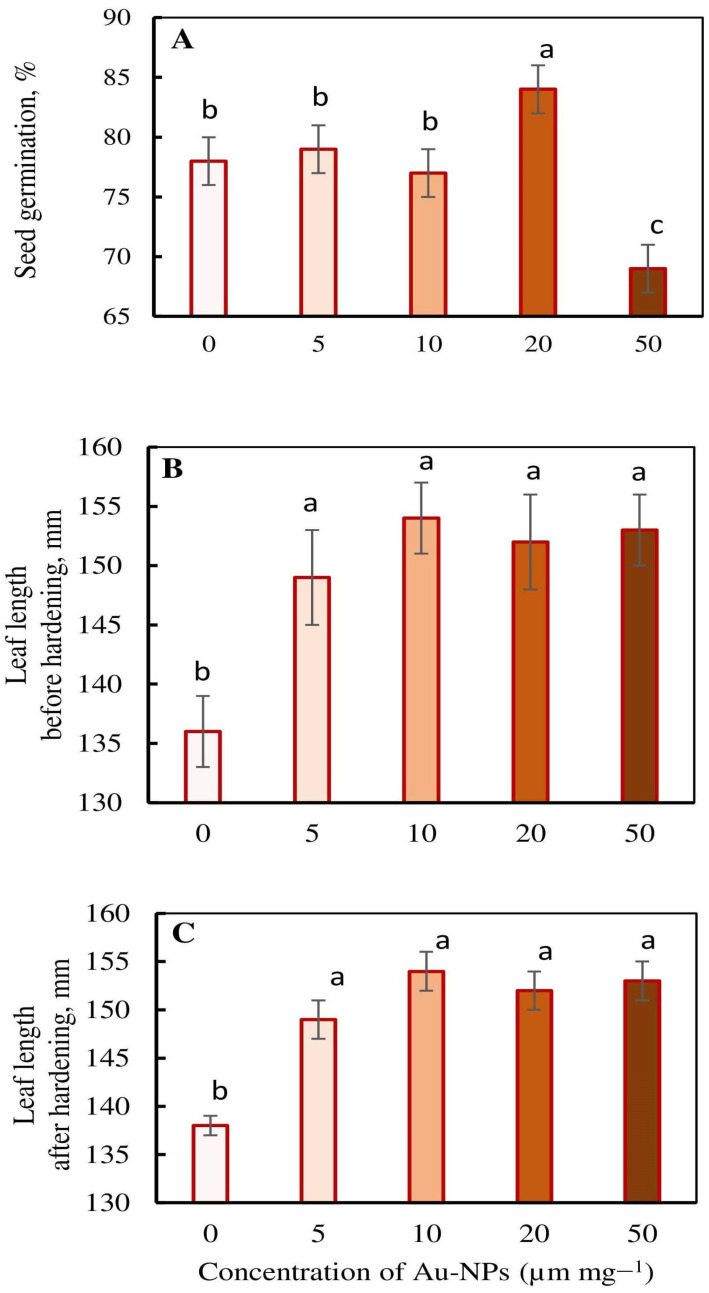
Impact of Au-based nanoparticles (Au-NPs) on seed germination (**A**) and leaf length before (**B**) and after (**C**) cold hardening (4 °C, 7 d) of wheat seedlings of genotype Zlata. In each variant of the treatment, 30 plants were used in 3 replications. The experiment was repeated 2 times. Different letters indicate mean values that are significantly different at *p* < 0.05.

**Figure 2 plants-13-01261-f002:**
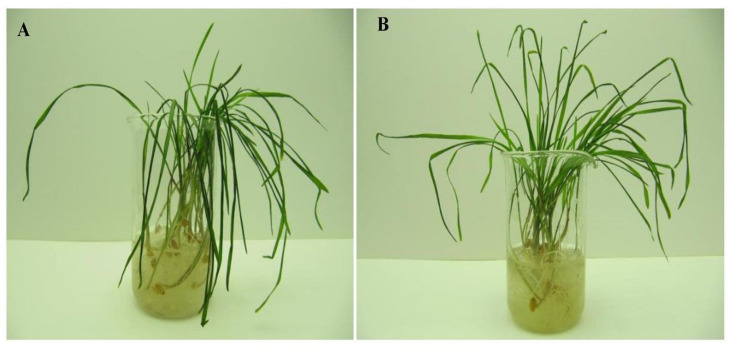
Unhardened wheat seedlings of genotype Zlata after freezing at −3 °C for 24 h: (**A**) control seedlings; (**B**) seedlings grown from seeds treated with Au-based nanoparticles at 10 µg mL^−1^.

**Figure 3 plants-13-01261-f003:**
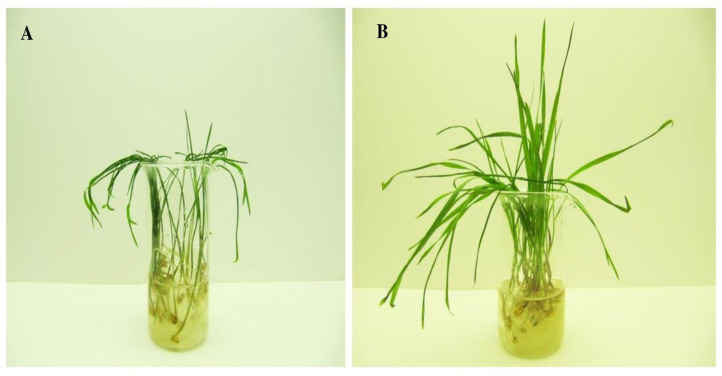
Hardened (4 °C, 7 d) wheat seedlings of genotype Zlata after freezing at −5 °C for 24 h: (**A**) control seedlings; (**B**) seedlings grown from seeds treated with Au-based nanoparticles at 10 µg mL^−1^.

**Figure 4 plants-13-01261-f004:**
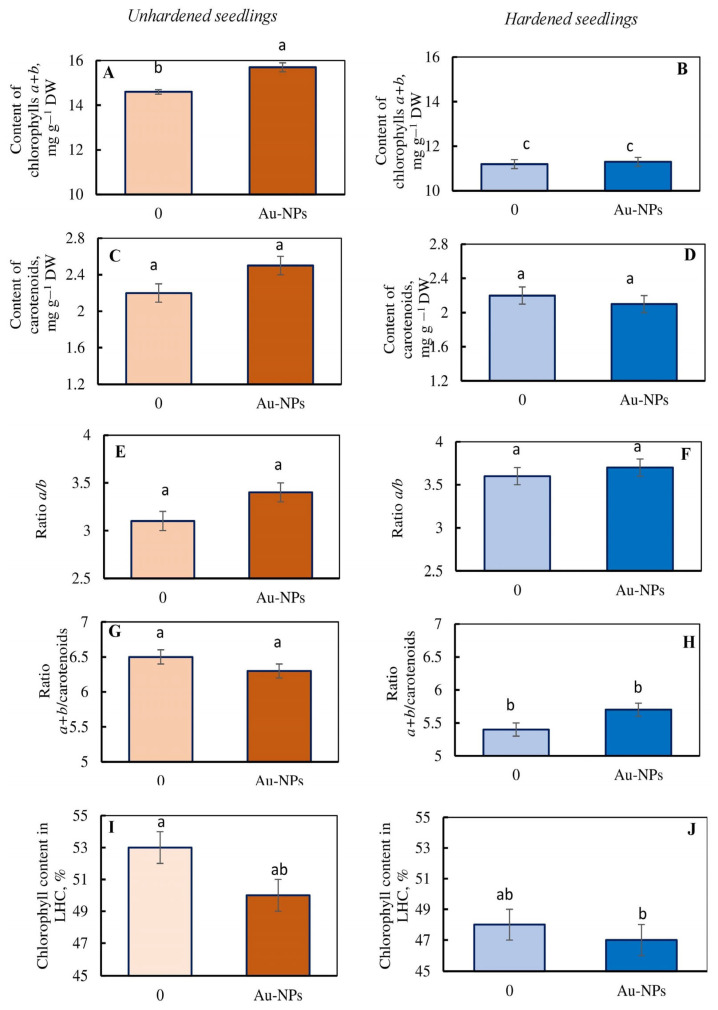
Impact of Au-based nanoparticles (Au-NPs) on content of chlorophylls *a + b* (**A**,**B**), carotenoids (**C**,**D**), ratio *a/b* (**E**,**F**) and *a + b*/carotenoids (**G**,**H**), and content of chlorophyll in light harvesting complex (LCH) (**I**,**J**) in leaves of unhardened (**A**,**C**,**E**,**G**,**I**) and hardened (4 °C, 7 d) (**B**,**D**,**F**,**H**,**J**) wheat seedlings of genotype Zlata. In each variant of the treatment 3 statistical repetitions were carried out. The experiment was repeated 3 times. Different letters indicate mean values, which are significantly different at *p* < 0.05. Au-NPs concentration—10 µg mL^−1^ (as concentrations causing maximum effect on tolerance to low temperature). DW—dry weight.

**Figure 5 plants-13-01261-f005:**
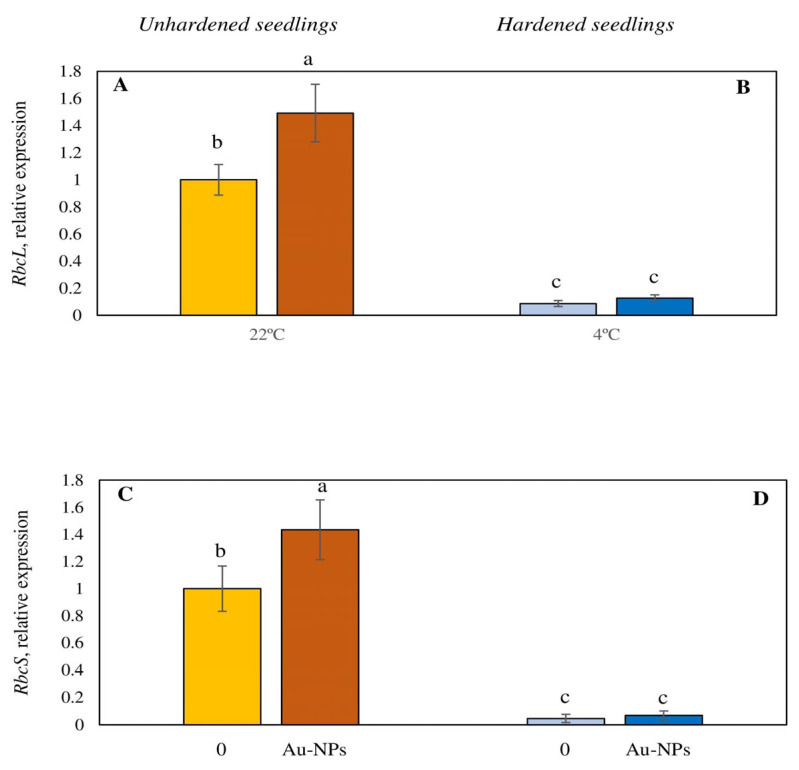
Impact of Au-based nanoparticles (Au-NPs) on *RbcL* (**A**,**B**) and *RbcS* (**C**,**D**) gene transcription of unhardened (**A**,**C**) and hardened (4 °C, 7 d) (**B**,**D**) wheat seedlings of genotype Zlata. In each variant of the treatment 3 statistical repetitions were carried out. The experiment was repeated 3 times. Different letters indicate mean values, which are significantly different at *p* < 0.05. Au-NPs concentration—10 µg mL^−1^ (as concentrations causing maximum effect on tolerance to low temperature).

**Figure 6 plants-13-01261-f006:**
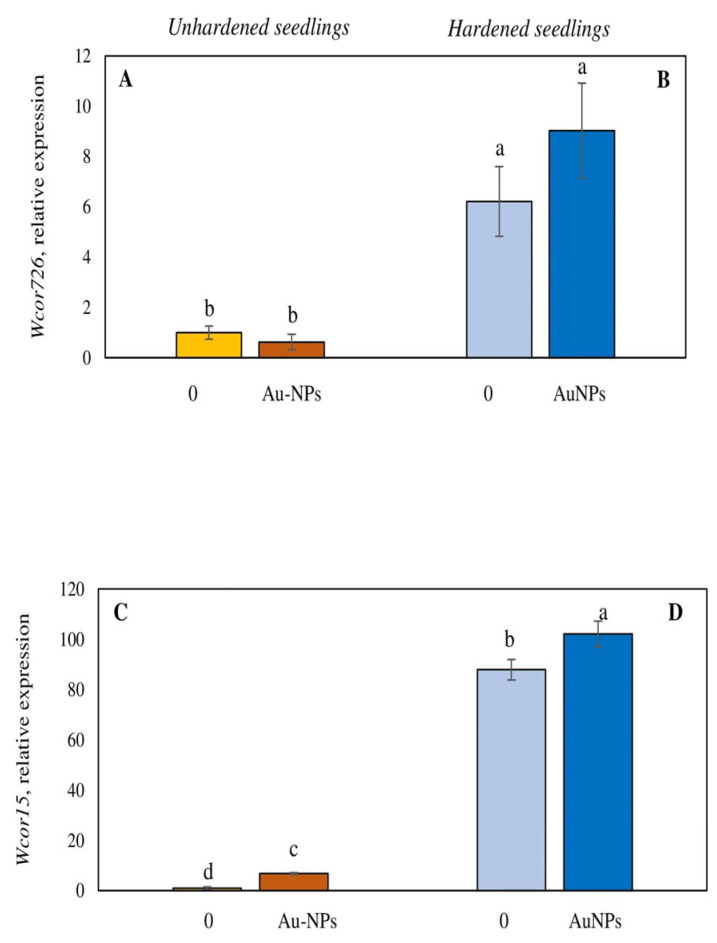
Impact of Au-based nanoparticles (Au-NPs) on *Wcor726* (**A**,**B**) and *Wcor15* (**C**,**D**) gene transcription of unhardened (**A**,**C**) and hardened (4 °C, 7 d) (**B**,**D**) wheat seedlings of genotype Zlata. In each variant of the treatment 3 statistical repetitions were carried out. The experiment was repeated 3 times. Different letters indicate mean values, which are significantly different at *p* < 0.05. Au-NPs concentration—10 µg mL^−1^ (as concentrations causing maximum effect on tolerance to low temperature).

**Figure 7 plants-13-01261-f007:**
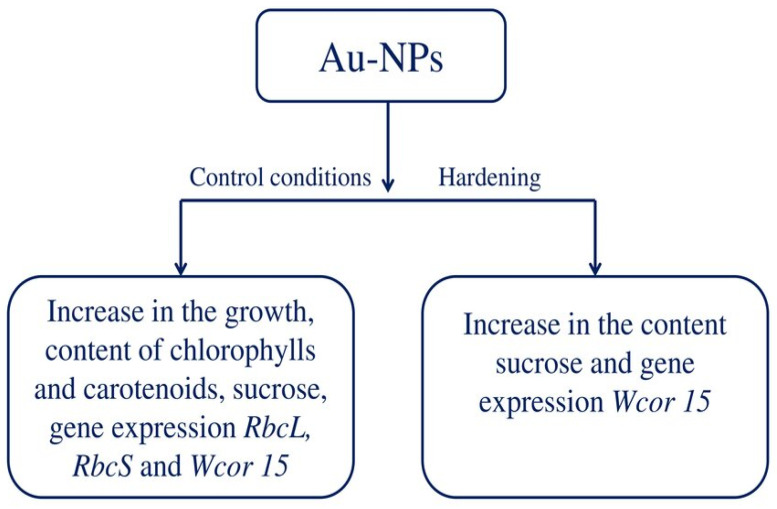
Some physiological, biochemical and molecular mechanisms of freezing tolerance improvement due to Au-NPs’ effects in seedlings of wheat of the cold-susceptible genotype Zlata.

**Figure 8 plants-13-01261-f008:**
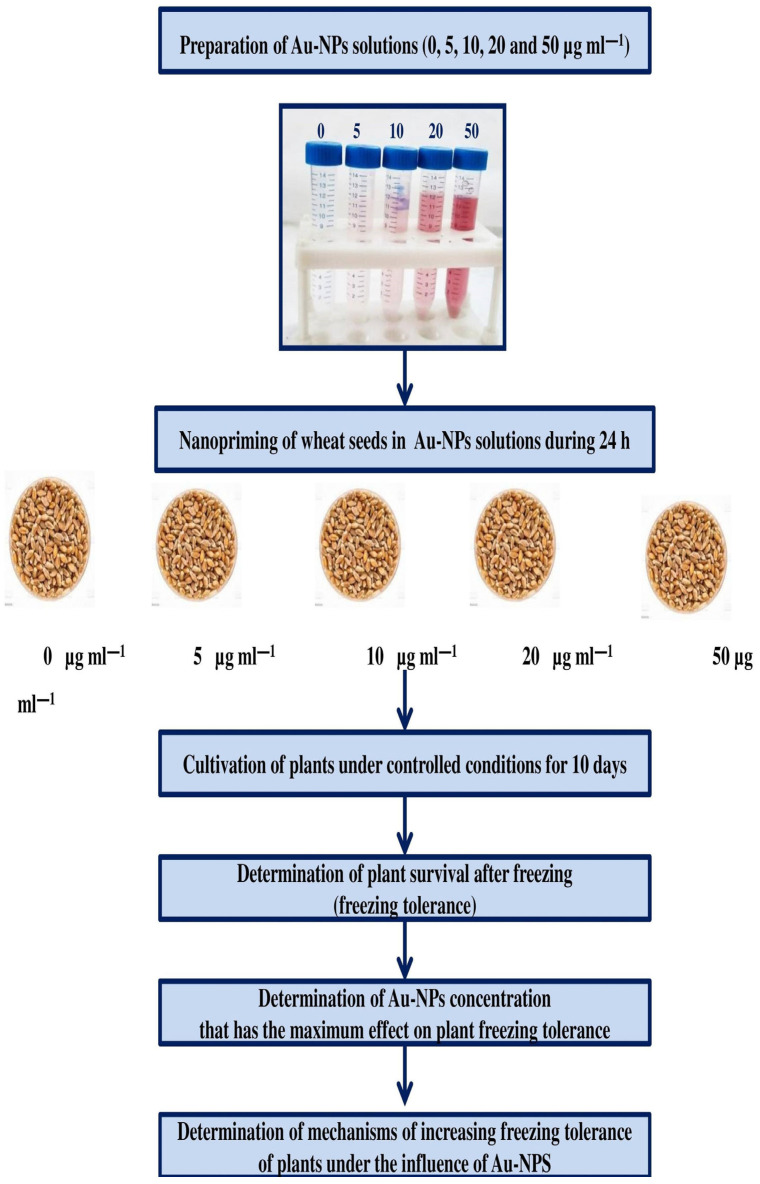
Design of experiment. Au-NPs—Au-based nanoparticles.

**Table 1 plants-13-01261-t001:** Impact of Au-based nanoparticles (Au-NPs) on the survival rate (%) of unhardened and hardened (4 °C, 7 d) wheat seedlings of genotype Zlata (concentration test).

Temperature of Freezing	Concentration of Au-NPs, µg mL^−1^				
	0	5	10	20	50
Unhardened seedlings					
−3 °C	15 ± 3 ^c^	30 ± 5 ^b^	50 ± 5 ^a^	10 ± 3 ^c^	10 ± 3 ^c^
−5 °C	0	0	0	0	0
Hardened seedlings					
−3 °C	60 ± 3 ^b^	90 ± 3 ^a^	97 ± 3 ^a^	90 ± 3 ^a^	90 ± 3 ^a^
−5 °C	7 ± 3 ^c^	50 ± 3 ^b^	67 ± 3 ^a^	50 ± 3 ^b^	50 ± 3 ^b^
−7 °C	0	5 ± 3 ^a^	7 ± 3 ^a^	6 ± 3 ^a^	0
−9 °C	0	0	0	0	0

Survival rate was calculated as the percentage of undamaged seedlings from the total number of plants frozen at −3, −5, −7 and −9 °C for 24 h. In each variant of the treatment, 30 plants were used in 3 replications (90 plants in all). The experiment was repeated 2 times. In every line the values that significantly differ at *p* < 0.05 are denoted by different letters.

**Table 2 plants-13-01261-t002:** Au concentrations in roots, seed and leaves of wheat seedlings of genotype Zlata in control variant and in plants treated with Au-based nanoparticles (Au-NPs).

Variant of Experiment	Au, µg g^−1^ Dry Weight		
	Roots	Leaves	Seeds
Control	<0.05	<0.05	<0.05
Au-NPs	1.6 ± 0.1	0.28 ± 0.01	3.9 ± 0.2

In each variant of the treatment, 3 statistical repetitions were carried out. The experiment was repeated 2 times. Au-NPs concentration—10 µg mL^−1^ (as concentrations causing the maximum effect on tolerance to low temperature).

**Table 3 plants-13-01261-t003:** Impact of Au-based nanoparticles (Au-NPs) on some physiological indicators of unhardened and hardened (4 °C, 7 d) wheat seedlings of genotype Zlata.

Variant of Experiment	MDA (µmol/g DW)	DW (%)	Monosaccharide (mg g^−1^ DW)	Sucrose (mg g^−1^ DW)	Sum of Sugars (mg g^−1^ DW)
Unhardened seedlings					
Control	36.8 ± 1.8 ^a^	13.2 ± 0.2 ^c^	55.0 ± 1.5 ^a^	4.4 ± 0.2 ^d^	59.4 ± 3.0 ^b^
Au-NPs	36.4 ± 1.2 ^a^	13.3 ± 0.2 ^c^	42.8 ± 1.5 ^b^	7.6 ± 0.2 ^c^	50.4 ± 2.5 ^c^
Hardened seedlings					
Control	20.8 ± 1.0 ^b^	17.8 ± 0.3 ^b^	45.2 ± 1.2 ^b^	14.1 ± 0.2 ^b^	59.3 ± 3.0 ^b^
Au-NPs	19.8 ± 1.3 ^b^	19.1 ± 0.4 ^a^	44.2 ± 1.7 ^b^	24.5 ± 0.2 ^a^	68.7 ± 3.4 ^a^

In each variant of the treatment 3 statistical repetitions were carried out. The experiment was repeated 3 times. Au-NPs concentration—10 µg mL^−1^ (as concentrations causing maximum effect on tolerance to low temperature). DW—dry weight. In each column the values that significantly differ at *p* < 0.05 are denoted by different letters.

**Table 4 plants-13-01261-t004:** Primers for qRT-PCR analysis.

Gene	Primer	Primer Sequences
*TaAct7*	Forward	TGCTATCCTTCGTTTGGACCTT
	Reverse	AGCGGTTGTTGTGAGGGAGT
*TaRP15*	Forward	TCATTGTGGAGGACTCGTGG
	Reverse	GCAGACATAGCCCACACAT
*RbcS*	Forward	GGATTCGACAACATGCGCCAGG
	Reverse	ATATGGCCTGTCGTGAGTGAGC
*RbcL*	Forward	ACCATTTATGCGCTGGAGAGACC
	Reverse	CAAGTAATGCCCCTTGATTTCACC
*Wcor726*	Forward	ACTGGAATGACCGGCTCG
	Reverse	TGTCCCGACTTCCCGTAGTT
*Wcor15*	Forward	CCACCCATCCATCAGCAGTT
	Reverse	CTTGGAGCGTTCTGCAGGC

## Data Availability

The datasets generated and/or analyzed during the current study are available from the corresponding author on reasonable request. The data are not publicly available due to ethical reasons.

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
