# Peer review of "Au-Based Nanoparticles Enhance Low Temperature Tolerance in Wheat by Regulating Some Physiological Parameters and Gene Expression"

_plants, 2024, doi:10.3390/plants13091261_

Round 1

Reviewer 1 Report

Comments and Suggestions for Authors

The manuscript presents a coherent objective and hypothesis. The methodology is clear. Check standards for citing units of quantities. I suggest at the end of the conclusion to insert a future recommendation for new insights.

Reviewer 2 Report

Comments and Suggestions for Authors

The paper reads well with informative introduction, well-described methodology and results. The reference section is sufficient.

The discussion section mentions “NPs in the environment”. The authors might like to include more detail: as NPs are beneficial in low concentrations, what might happen to environmental concentrations with intense use? Are there ways to recycle or dispose of NPs? 

Reviewer 3 Report

Comments and Suggestions for Authors

Review report for manuscript ID plants-2940296 entitled ‘Au-based Nanoparticles Enhance Low Temperature Tolerance in Wheat by Modifying Some Physiological Parameters, and Regulating Gene Expression Related to Photosynthetic Apparatus and Cold Response

 Summary and opinion

The manuscript presents data on the use of Au-based nanoparticles to enhance freezing tolerance of a cold sensitive wheat cultivar. Seed were soaked in nanoparticles solutions and the effectc of the treatment on cold tolerance and physiological parameters related to cold tolerance were studied. The use of nanoparticles enhanced frost tolerance of wheat seedlings and influenced parameters like sucrose content and gene expression.

 The manuscript is in general well prepared, and all figures are clear. However, the method for the freezing test as described in the methods is not ideal: it seems that seedlings were directly transferred to a freezing chamber at the different exposure temperatures for 24 hours. It would be much better using a cooling profile with a specific cooling rate (2-3 degrees/hour), followed by a defined exposure time and then followed again by a specific thawing rate.

The mechanisms how nanoparticles influence freezing stress are not clear. It’s very likely that the presented changes in sugar content and gene expression are responsible for it. The discussion should address this more carefully, I think its too early to recommend the use of nanoparticles as stress adaptogen in plants.

General comments:

The title is very descriptive and quite long, it could be more focused on the results.

Specific comments

Line 22: in my opinion it’s too early to recommend the use of nanoparticles for frost defense. Results sound promising but there is no evidence about the use of nanoparticles directly in the field and how long the effects on freezing stress will last

Line 117: “most of the plant died” survival rate is 0 at -5°C, so all the plants died?

Line 120: “7% to 60%” the presented data show values from 7-67%

Table 1: as mentioned above the freezing protocol is not ideal. However, the effects are very clear. But it would be better to adapt exposure temperatures in a way that also initial damage is measured. The chosen temperature of -3°C in unhardened seedlings caused already a damage of 85%. Exposure temperature of -1°C or -2 °C probably would show also initial frost damage.

What are the causes of freezing damage in wheat plants? Ice formation in the tissue? The causes are not discussed. Could there also be a effect of nanoparticles on ice nucleation temperatures?

Figures 4-6:differences are much more pronounced between unhardened and hardened seedlings (which can be expected). The effects of nanoparticle treatments are less pronounced, anyway significant differences were found in some parameters. As the underlying mechanisms how nanoparticles act in plant tissues are not clear, this has to be stated mor carefully in the discussion

Line 520: in my opinion its too early for recommendations

Comments on the Quality of English Language

Minor editing required

Round 2

Reviewer 3 Report

Comments and Suggestions for Authors

Manuscript Number:  plants-2940296

Title: Au-based Nanoparticles Enhance Low Temperature Tolerance in Wheat by Regulating Some Physiological Parameters and Gene Expression

The manuscript was improved and all the comments were adressed. 

Comments on the Quality of English Language

Minor editing of English language required